# Enhancing Adaptive History Reserving by Spiking Convolutional Block Attention Module in Recurrent Neural Networks

Qi Xu[1]    Yuyuan Gao[1]    Jiangrong Shen[2*]    Yaxin Li[1]
Xuming Ran[1]    Huajin Tang[2]    Gang Pan[2]

[1]School of Artificial Intelligence, Dalian University of Technology
[2]College of Computer Science and Technology, Zhejiang University

## Abstract

Spiking neural networks (SNNs) serve as one type of efficient model to process spatio-temporal patterns in time series, such as the Address-Event Representation data collected from Dynamic Vision Sensor (DVS). Although convolutional SNNs have achieved remarkable performance on these AER datasets, benefiting from the predominant spatial feature extraction ability of convolutional structure, they ignore temporal features related to sequential time points. In this paper, we develop a recurrent spiking neural network (RSNN) model embedded with an advanced spiking convolutional block attention module (SCBAM) component to combine both spatial and temporal features of spatio-temporal patterns. It invokes the history information in spatial and temporal channels adaptively through SCBAM, which brings the advantages of efficient memory calling and history redundancy elimination. The performance of our model was evaluated in DVS128-Gesture dataset and other time-series datasets. The experimental results show that the proposed SRNN-SCBAM model makes better use of the history information in spatial and temporal dimensions with less memory space, and achieves higher accuracy compared to other models.

## 1   Introduction

Neurons in the human brain maintain preceding activity traces on the molecular level [1, 2]. Especially, the eligibility traces are induced by the fading memory of spike events where the presynaptic neuron fired before the postsynapstic neuron [3]. Inspired by the loop behaviors of neurons, Recurrent spiking neural networks (RSNNs), have been proposed and proven to have potential in processing spatio-temporal patterns through learning from the historical context. Many of those existing approaches suggest memory preserving and extraction mechanisms play a crucial role in exploiting historical context underlying spatio-temporal dynamics. Although the current recurrent structures of RSNNs can capture the spatial and temporal variations at each level of the networks, the importance of different channels at different time points is ignored [4–6] which contributes the memory preservation and extraction can not focus on the small and important channels at specific time points.

Different recurrent structures of SNNs have been proposed to capture the memory. Like Long Short-Term Memory (LSTM) cells in ANNs, the LSTM spiking networks [7] preserve valuable historical information by latching the gradients of a hidden state, proficient in processing time series data. To enhance the spatial feature extraction ability further of LSTM, its extension, named ConvLSTM, with

---

*Corresponding authors: jrshen@zju.edu.cn.

convolutional structures in both input-to-state and state-to-state transitions. Meanwhile, the Address-event representation (AER) [8, 9] data is one typical spatio-temporal pattern and has contextual spatial and temporal cues. Considering the advantages of ConvLSTM over LSTM, we aim to introduce the ConvLSTM structure into SNNs, to construct the recurrent spiking networks with ConvLSTM structures.

The core benefit of SNNs with recurrent structures is the memory invoking along with sequential time points. However, most of the current recurrent structures of SNNs ignore the influence distinction of different historical contexts on spatial and temporal dimensions. The attention mechanism has been studied and achieved to tell where to focus and improve the representation of interests. The convolutional block attention module (CBAM) extracts the key channel and spatial information by emphasizing the significant features. We adopt the CBAM component to implement the history invoking adaptively in forgetting gates in RSNNs.Because CBAM can conduct Attention on the channel and spatial dimensions. Because CBAM can conduct Attention on the channel and spatial dimensions, we can adjust the weight of forgetting gate information. Therefore, we propose the spiking ConvLSTM with Spiking CBAM component to invoke the history information in spatial and temporal channels adaptively, which brings the advantages of efficient memory calling and history redundancy elimination.

The contributions of the paper are as follows:

- The spiking recurrent neural networks model with the spiking convolutional block attention module (CBAM) component (SRNN-SCBAM) is proposed to combine both spatial and temporal features of spatio-temporal patterns.

- The proposed model invokes the history information in spatial and temporal channels adaptively through spiking CBAM, which brings the advantages of efficient memory calling and history redundancy elimination.

- The experiments are conducted on CIFAR10-DVS and DVS128-Gesture datasets. The ablation study and comparison results show that our model achieves competitive performance among current RSNNs models.

## 2 Related Work

**Recurrent SNNs.** Recurrent structures have manifested great potential in processing spatio-temporal patterns, benefiting from the information integration over time by multiple participant computations of each recurrent neuron. Current RSNNs contain two different kinds of network architectures according to the structural constraints. Firstly, based on the reservoir with randomly connected recurrent structures, The first-order reduced and controlled error algorithm could learn arbitrary firing patterns by stabilizing strong chaotic rate fluctuations in a network of excitatory and inhibitory neurons respecting Dale's law. Considering the inefficient and unstable connectivity of separating the excitatory and inhibitory populations, the framework of training the continuous-variable rate RSNNs with biophysical constraints and transferring the learned dynamics to spiking RSNNs, by introducing one parameter to establish the equivalence between rate and spiking RSNNs [10, 11].

Another category of the RSNNs is based on learnable recurrent structures. Learning from the training method of artificial neural networks with recurrent structures, the backpropagation through time (BPTT) is applied to the training of RSNNs [12, 13], which requires propagation of gradients backward in time by unfolding recurrent layers. In the view of biologically realistic [14, 15], e-prop optimally combines the local eligibility traces and top-down signals and formulates the gradient descent learning in RSNNs [16]. Besides, RSNN models can also be implemented on neuromorphic hardware, such as SpiNNaker [17] or Loihi chips [18], bringing computing efficiency combined with device properties.

**Attention Mechanism.** Human's visual system exploits the sequence of partial glimpses instead of processing the whole scene at once. Inspired by that property, the attention mechanism concentrates on the selective information in the input according to the sensitivity of the output to the variant inputs [19–21]. [22] propose temporal-wise attention SNN (TA-SNN) model to select relevant frames in the training stage and discard irrelevant frames in the inference stage by judging the significance of frames for the final decision. The convolutional block attention module (CBAM) module has been applied to CNNs to capture meaningful features along spatial and channel dimensions. The

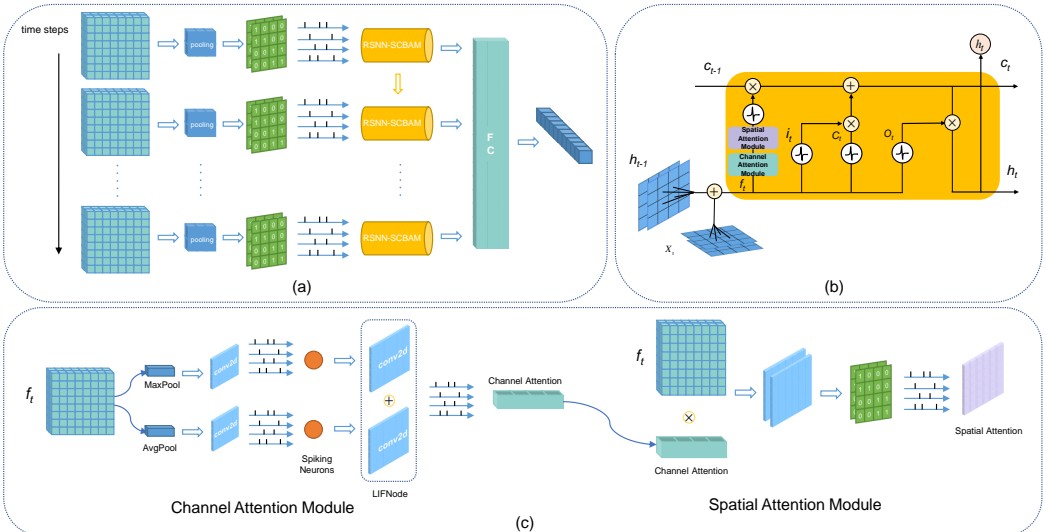

Figure 1: The framework of the proposed RSNN-SCBAM. (a) The network structure of the RSNNs with spiking ConvLSTMs and fully-connected layers. The spiking CBAM can adaptively select the key features both in spatial and temporal domains, and its effect on forgetting gating of spiking ConvLSTMs can invoke the history memories efficiently and eliminate the history redundancy, thus improving the processing of the spatiotemporal patterns. (b) The spiking ConvLSTMs are implemented to exploit the spatiotemporal feature through the proposed spiking convolutional component and spiking LSTM module. (c) The spiking CBAM component captures the sparse and complementary key features on the spatial and channel domain.

intermediate feature map in CNNs is adaptively refined through CBAM at each convolutional block [23]. However, there is still a lack of combination attention modules to apply in RSNNs both in temporal and spatial dimensions.

## 3 Method

In this section, we show detailed information on the SRNN-SCBAM model. In order to solve the problem that most SNN recurrent structures ignore the difference in the impact of different historical backgrounds on the spatial and temporal dimensions, we propose that the spiking ConvLSTM with CBAM components can adaptively call the historical information in the spatial and temporal channels. We can model DVS and other time series data, and complete the tasks of classification and prediction. For other data sets such as cifar10DVS, we additionally add convolution pooling to deepen the network structure.

### 3.1 The framework of the RSNN-CBAM

In the classification task for DVS128 Gesture, we used the network framework shown in Figure 1. The entire network of RSNN-CBAM consists of a layer of pooling layer, a layer of Spiking Convlstm with CBAM components, and a fully connected layer. Using SRNN, a PyTorch-based encoder-classifier model is implemented for neuromorphic data sets such as DVS128 Gesture. sort. Use pooling to reduce the dimensionality of data and remove redundant information, use Spiking Convlstm with CBAM components to extract spatial and channel features from neuromorphic data, and then send them to the classifier for classification.

### 3.2 Spiking ConvLSTM

We introduce ConvLSTM into SNN and use its time and space feature extraction ability to accurately process time series data. Spiking ConvLSTM is similar to Ann ConvLSTM, and it also overcomes the shortcomings of LSTM in processing three-dimensional information. And like the traditional

LSTM, the core concept of spiking ConvLSTM lies in the cell state, $c_t$, and "gate" structure. The cell state is able to pass on relevant information in the process of sequence processing all the way. The "gate" structure learns which information to keep or forget during training.

Spiking ConvLSTM input is a 3D tensor, and the last two dimensions are spatial dimensions (Height and Weight). For the data at each timestep T, Spiking ConvLSTM replaces a part of the connection operation in LSTM with a convolution operation, which has a convolution structure in both input-to-state and state-to-state transitions, through the current input and the past state of local neighbors.

Spiking ConvLSTM is more suitable for processing dynamic spatio temporal patterns, such as gesture recognition with clear gesture moving trajectories but without preserving the details of fine-grained images with high-resolution pixels.

More specifically, given a set of spiking inputs $x_1, x_2, , x_T$, the gates and states are characterized as follows:

$$
\begin{aligned}
\boldsymbol{f}_t &= \sigma_1\left(w_{f,h} * \boldsymbol{h}_{t-1} + w_{f,x} * \boldsymbol{x}_t + \boldsymbol{b}_f\right), \\
\boldsymbol{i}_t &= \sigma_1\left(w_{i,h} * \boldsymbol{h}_{t-1} + w_{i,x} * \boldsymbol{x}_t + \boldsymbol{b}_i\right), \\
\boldsymbol{g}_t &= \sigma_2\left(w_{g,h} * \boldsymbol{h}_{t-1} + w_{g,x} * \boldsymbol{x}_t + \boldsymbol{b}_g\right), \\
\boldsymbol{c}_t &= \boldsymbol{f}_t \odot \boldsymbol{c}_{t-1} + \boldsymbol{i}_t \odot \boldsymbol{g}_t, \\
\boldsymbol{o}_t &= \sigma_1\left(w_{o,h} * \boldsymbol{h}_{t-1} + w_{o,x} * \boldsymbol{x}_t + \boldsymbol{b}_o\right), \\
\boldsymbol{h}_t &= \boldsymbol{o}_t \odot \boldsymbol{c}_t,
\end{aligned}
\tag{1}
$$

where $\odot$ represents the Hadamard product,* represents convolution operation, and $\sigma_1$ and $\sigma_2$ are spike activations that map the membrane potential of a neuron, to a spike if it exceeds the threshold value respectively. Also, w, and b, denote associated weights and biases. For the network, respectively. $x_t$ and $h_{t-1}$ respectively correspond to the spiking input at the current moment and the hidden state at the previous moment.

The forget gate, denoted by $f_t$, decides which information should be ignored. The input gate $i_t$, which controls the information entering the cell, has another auxiliary input layer $g_t$, modulated by another peak activation $sigma_2$. Finally, the output of the spiking Convlstm is formed based on the output gate $o_t$ and the cell state $c_t$.

### 3.3 Spiking CBAM

On the basis of spiking Convlstm, we use the attention mechanism to improve the forget gate. The spiking ConvLSTM with CBAM component is proposed to adaptively call historical information in space and time channels. The forget gate in spiking Convlstm is obtained by splicing the pulse input at the current moment and the hidden state at the previous moment, and then obtaining it through convolution and segmentation. For the CBAM component in RSNN-CBAM, the forget gate is used as input, and it will sequentially extract the one-dimensional channel attention map and the two-dimensional space attention map of the forget gate, thereby extracting the channel features and spatial features of the forget gate, and further, for space and time, the historical information in the channel is adjusted. The overall attention process can be summarized as:

$$
\begin{aligned}
\mathbf{f}_t' &= \mathbf{M_c}(\mathbf{f}_t) \otimes \mathbf{f}_t, \\
\mathbf{f}_t'' &= \mathbf{M_s}\left(\mathbf{f}_t'\right) \otimes \mathbf{f}_t',
\end{aligned}
\tag{2}
$$

Where $\otimes$ denotes element-wise multiplication. $\mathbf{M}_c$ is the channel attention module, and $\mathbf{M}_s$ is the spatial attention module. Mc and Ms sequentially perform channel feature extraction and spatial feature extraction on the forget gate to generate channel attention module $\mathbf{M_c}(\mathbf{f}_t)$ and spatial attention module $\mathbf{M_s}(\mathbf{f}_t')$. $\mathbf{f}_t''$ is the spiking CBAM output. The calculation process of each attention diagram is shown in the figure. The following describes the details of each attention module.

The channel attention component in Spiking CBAM uses balanced pooling and maximum pooling features to generate a one-dimensional channel attention map of forget gate through SNN-MLP. The channel attention is calculated as:

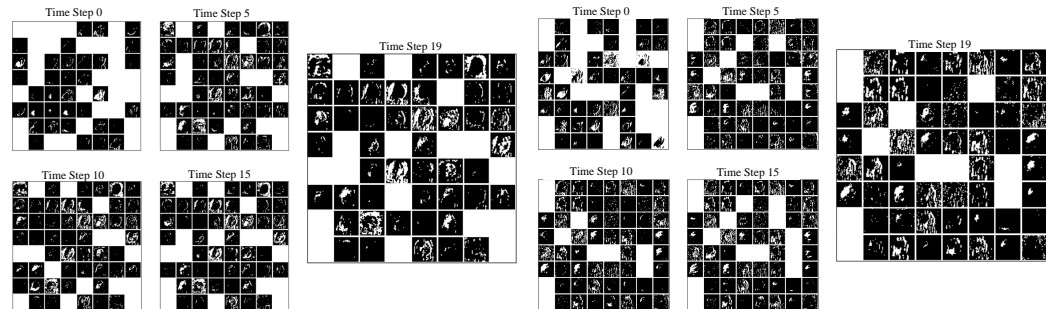

Figure 2: The visualization of extract features of RSNNs with and without SCBAM. We draw the feature map in spiking Convlstm layer at the time steps of $0_{th}$, $5_{th}$, $10_{th}$, $15_{th}$, and $19_{th}$. The RSNNs with spiking CBAM capture more sparse and complementary features compared with the RSNNs without spiking CBAM.

$$\mathbf{M_c}(\mathbf{f_t}) = \sigma(\mathrm{SMLP}(\mathrm{AvgPool}(\mathbf{f_t})) + \mathrm{SMLP}(\mathrm{MaxPool}(\mathbf{f_t}))), \tag{3}$$

where $\sigma$ represents the number of spiking activations, Average pooling, and maximum pooling feature map share SNN-MLP weight.

The spatial attention component in Spiking CBAM applies the average pooling and maximum pooling operations along the channel axis, performs the convolution operation on it, and sends it into the spiking activation function to generate the two-dimensional spatial attention feature map of the forget gate. The spatial attention is calculated as:

$$\mathbf{M_s}(\mathbf{f_t'}) = \sigma\left(f^{7\times7}([\mathrm{AvgPool}(\mathbf{f_t'}); \mathrm{MaxPool}(\mathbf{f_t'})])\right), \tag{4}$$

Where $\sigma$ represents the number of spiking activations, and $f^{7\times7}$ represents a convolution operation with the filter size of $7 \times 7$ [23].

Through the above two attention mechanisms, we obtained the one-dimensional channel attention map and two-dimensional spatial attention map of the forget gate, extracted the channel features and spatial features of the forget gate, and used Spiking CBAM adaptively recalls historical information in both spatial and temporal channels, and ablation studies and visualizations are performed to show the effectiveness of the attention component.

### 3.4 Surrogate Gradient Method

We mainly use the Leaky integrate-and-fire (LIF) node as the basic unit. The LIF model is also the most commonly used model to describe the dynamics of SNN neurons, which can be described by the following equation:

$$\tau\frac{du(t)}{dt} = -u(t) + I(t), \tag{5}$$

Where $u(t)$ is the neuron membrane potential (membrane potential) at time t, $\tau$ is the time constant, and $I(t)$ represents the presynaptic signal determined by the previous neuron's pulse activity or external stimulus and synaptic weights. enter. When the membrane potential u exceeds a specified threshold the neuron fires a pulse and resets its membrane potential to $u_{reset}$.

In order to solve the problem that the loss function is not differentiable in the backpropagation in SNN, we use the surrogate gradient training method [24–26]. The substitution function is used to generate spikes in spiking Convlstm and spiking CBAM.

The principle of the gradient substitution method is to use the Heaviside step function in the forward propagation: $y = \theta(x)$, While backpropagating, the substitution function is used: $dy/dx = \sigma(x)'$. Where $\sigma(x)$ is a function similar in shape to $\theta(x)$ but smooth and continuous. We mainly use the

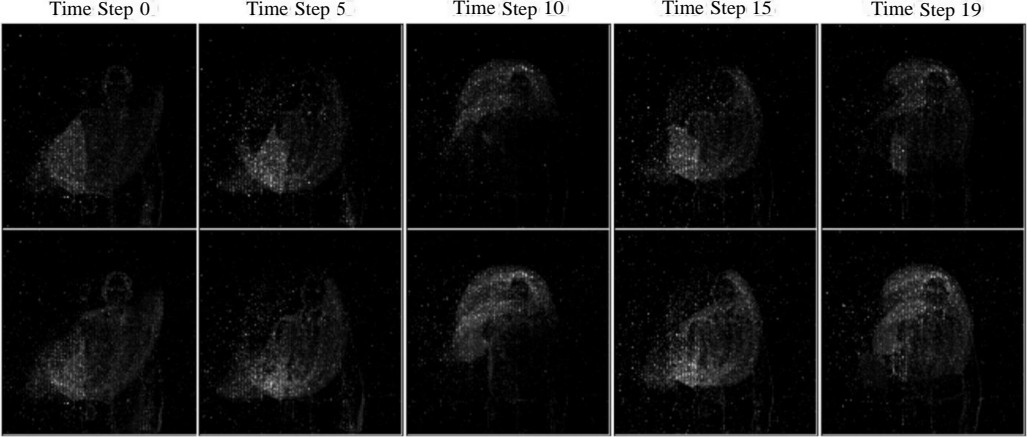

Figure 3: Visualization of input neuromorphic data with a size of 2*128*128. We plot the feature maps of the input image at time steps of $0_{th}$, $5_{th}$, $10_{th}$, $15_{th}$, and $19_{th}$.

spike function of the gradient of the Gaussian error function (erf) during backpropagation. The corresponding original function is:

$$g(x) = \frac{1}{2}(1 - \mathrm{erf}(-\alpha x)) = \frac{1}{\sqrt{\pi}} \int_{-\infty}^{\alpha x} e^{-t^2} dt, \tag{6}$$

Where $\alpha$ is the parameter that controls the smoothness of the gradient during backpropagation. Backpropagation employs the gradient descent rule according to the derivatives:

$$g'(x) = \frac{\alpha}{\sqrt{\pi}} e^{-\alpha^2 x^2}, \tag{7}$$

Then, the derivatives of the gradient of the variables can be approximated by the above formulations.$\alpha$ is the derivatives of the gradient of the variables can be approximated by the above formulations.

## 4 Results

We conduct the following experiments to evaluate the performance of the proposed RSNN-CBAM models. We first introduce the experimental settings. Then, the classification accuracy of RSNN-CBAM is recorded and compared to the current state-of-the-art models. The ablation study and visualization are followed to indicate the effectiveness of the attention component.

### 4.1 Experimental Setup

In order to further show the performance of the proposed method on dynamic spatio-temporal information processing, we evaluate the classification accuracy on the CIFAR10-DVS and DVS128 Gesture datasets. CIFAR10-DVS is an event-based version of the classical CIFAR10 dataset which also contains 10 classes with 10,000 event streams converted from images. Similar to CIFAR10-DVS, DVS128-Gesture is an event-based spiking dataset that contains 11 hand gestures from 29 subjects. Based on these spatio-temporal event streams [27, 28], the proposed method was evaluated for dynamic information processing and its adaptive history reserving ability.

Detailed parameter settings are listed in Table 1. The learning rate scheduler for RSNN-CBAM is set to be 1e-3 (selected from linear warm-up and decay or step decay). The learning epoch is 200 and the batch size is 32. The integration time constants $\tau$ of LIF neurons are set to be 4.0. $\alpha$ is the parameter that controls the smoothness of the gradient during backpropagation for surrogate gradient learning process, which is set to be 4.0 in this paper. $\boldsymbol{u}_{reset}$ is the reset voltage of the neuron and set to be 0. The above content will be added to the revision.

Table 1: Experimental parameter settings.

| Parameters | DVS Gesture | Cifar10-DVS |
|---|---|---|
| Epoch | 200 | 1024 |
| Batchsize | 32 | 32 |
| Learning rate | 1e-3 | 1e-3 |
| LIF, $\tau u$ | 2.0 | 2.0 |
| $\alpha$ | 4.0 | 4.0 |
| $u_{reset}$ | 0 | 0 |

Table 2: Accuracy of different solutions for the DVS128 GESTURE dataset (11 classes).

| Method | f | i | o | Accuracy (%) |
|---|---|---|---|---|
| Spiking Convlstm | 0 | 0 | 0 | 93.3 |
| spiking Convlstm+CBAM | 1 | 0 | 0 | 92.4 |
| Spiking Convlstm+Spiking CBAM | 0 | 1 | 1 | 86.1 |
| Spiking Convlstm+Spiking CBAM | 1 | 1 | 1 | 88.8 |
| Spiking Convlstm+Spiking CBAM | 1 | 0 | 0 | 95.5 |
| Spiking Convlstm+Spiking CBAM | 1 | 1 | 0 | 92.0 |
| Spiking Convlstm+Spiking CBAM | 0 | 1 | 0 | 91.3 |

ALL the following experiments were conducted on either an A100-SMX GPU. The network structure is composed of one layer of Convlstm neurons and three FCLIF1D layers. The results of the experiments are given in Table 2, as shown in this table, most of the accuracies among the various structures are increased based on the proposed method.

## 4.2 Ablation Study

In this section, the ablation study is conducted to validate the effectiveness of each component proposed in this paper.

**The effectiveness of SCBAM in SRNN.** The spiking CBAM component is employed in SRNN to implement selective memory preservation. As illustrated in Tab. 2, we conduct the ablation study on the DVS128 GESTURE dataset. Firstly, the Spiking ConvLSTM with spiking CBAM module achieves the highest test accuracy of 95%, which performs better than the one without any CBAM component over a 1.7% accuracy gap. It indicates the effectiveness of Spiking CBAM in improving the performance of Spiking ConvLSTM. Secondly, the proposed Spiking CBAM component achieves better classification accuracy than the accuracy drop of 2.6% original CBAM component. It suggests that the proposed spiking CBAM component captures more precise features than the original CBAM in the spiking ConvLSTM module because of its better temporal feature selection capability.

**Explore the adaption of different gates to SCBAM.** We employ the spiking CBAM on the forgetting gate to control the memory preserving and invoking for Spiking ConvLSTM. Here we further explore the performance of other gates in Spiking ConvLSTM to show the efficiency of the forgetting gate. As shown in the last five columns in Table. 2, the results suggest the priority of the forgetting gate compared with the other two input and output gates. The forgetting gate obtains the largest accuracy and exceeds nearly 10% accuracy than other gatings. It is reasonable that the forgetting gate with spiking CBAM controls the history information invoking and produces a marked effect on the effective computation of spiking ConvLSTM.

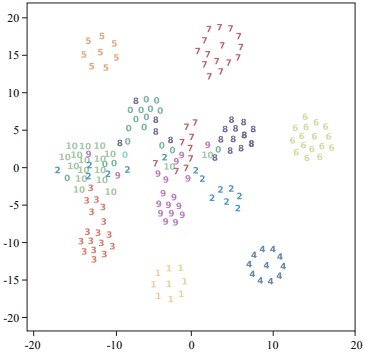
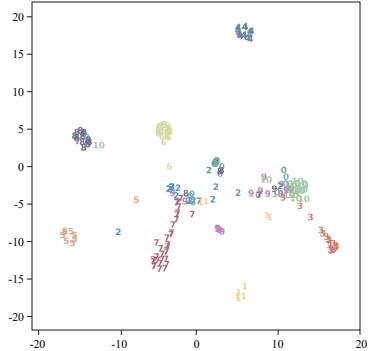

(a) The visualization of extract features of RSNNs with SCBAM

(b) The visualization of extract features of RSNNs without SCBAM

Figure 4: The visualization of extract features of RSNNs with and without SCBAM. The RSNNs with spiking CBAM capture more sparse and complementary features compared with the RSNNs without spiking CBAM.

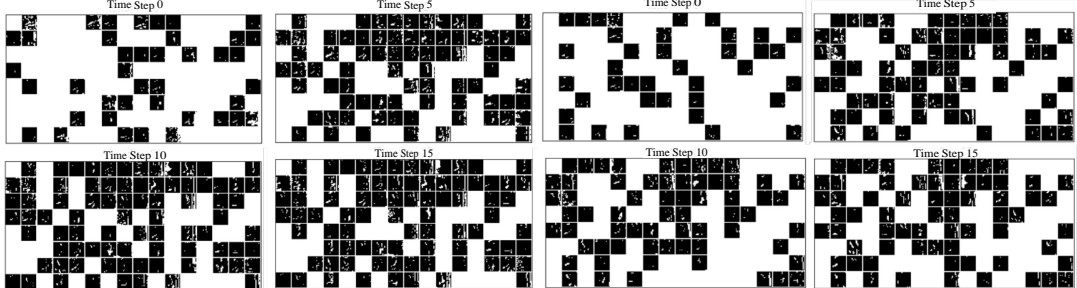

(a) The output of RSNN with Spiking CBAM          (b) The output of RSNN without Spiking CBAM

Figure 5: The figure above depicts visualizations of RSNN output tensors on the CIFAR10-DVS dataset. We selected time steps of 0, 5, 10, and 15 to compare the extracted features of RSNNs with and without SCBAM. It shows that the RSNN module with SCBAM captures information as anticipated, exhibiting strong sparsity across both temporal and spatial dimensions throughout the entire time steps while the RSNN module without SCBAM only extracts limited features which may lose a significant amount of crucial information.

## 4.3 Visualization analysis

To validate the effect of the SCBAM on the selection of the key features in the spatial and channel dimensions, we observe the feature map visualization of the RSNN-SCBAM model on the DVS128 GESTURE and CIFAR10-DVS datasets under the situation with and without SCBAM component.

DVS128 GESTURE feature map visualization results are illustrated in Fig. 4, the RSNN module with SCBAM captures more complementary and sparse features than the one without SCBAM. On one hand, the SCBAM possesses fine sparsity in the channel and spatial dimensions through the whole time steps. That benefits from the attention mechanism that captures significant features and eliminates redundant features. On the other hand, most of the features extracted by SCBAM are complementary to each other. Each feature map focuses on one specific area such as the hands and the shoulders. However, some feature maps of the RSNNs without SCBAM are quite similar because of the lack of filtering of redundant information. Based on the proposed SRNN-CBAM, the historical redundancy left in other spiking recurrent models could be discarded sharply, meanwhile, the proposed method maintains good performance by adaptive history reserving.

CIFAR10-DVS feature map visualization results are illustrated in Fig. 5, we visualize the extracted features at several time steps of the proposed RSNNs with and without SCBAM. It shows that the RSNN module with SCBAM captures information as anticipated, exhibiting strong sparsity across both temporal and spatial dimensions throughout the entire time steps while the RSNN module without SCBAM only extracts limited features which may lose a significant amount of crucial information.

We further conducted ablation experiments on dynamic thresholding during training on the DVSGesture dataset, and our RSNN-SCBAM model adopted two different dynamic thresholding methods in LIFNode and MultiStepLIFNode SNN. We set the thresholds of MultiStepLIFNode and LIFNode as trainable parameters and conduct experiments to verify their impact. The experimental results are shown in Table. 3 shown.

Table 3: Accuracy of two different dynamic threshold methods for the DVS128 GESTURE dataset.

| Method | Node_threshold | Accuracy (%) |
|---|---|---|
| RSNN-SCBAM | **LIFNode** | **91.3** |
| RSNN-SCBAM | **MultiStepLIFNode** | **89.9** |

Table 4: Accuracy of additional time step values of 10, 15, and 25 on the DVSGesture dataset.

| Method | Time step | Accuracy (%) |
|---|---|---|
| RSNN-SCBAM | 10 | **89.2** |
| RSNN-SCBAM | 15 | **92.4** |
| RSNN-SCBAM | 25 | **90.3** |

However, the current results do not show any significant improvement in dynamic thresholds. This threshold adjustment may make the model relatively complex, involving more parameters and

Table 5: Accuracy of solutions for the CIFAR10-DVS dataset (10 classes).

| Method | Accuracy (%) |
|---|---|
| HOTS[29] | 27.1 |
| Lightweight Statistical[30] | 31.2 |
| Attention Mechanisms[31] | 44.1 |
| HATS[32] | 52.4 |
| LIF-NET[33] | 63.5 |
| LIAF-NET[33] | 70.4 |
| Proposed CBAM-SNN | **66.3** |

nonlinear operations. If the parameters or network structure of the model are not adjusted correctly, poor model performance may result. Furthermore, SNNs may require large amounts of training data and longer training times to achieve effective learning and optimization.

Regarding the variation in LSTM time steps, we conduct experiments using additional time step values of 10, 15, and 25 on the DVSGesture dataset. As shown in Table. 4, the results show that the time step of 20 is the optimal choice. This could be attributed to the nature of the DVSGesture data, where gestures are repeated by individuals at specific time intervals. The accuracies of the RSNN-CSAM increase first and then reduce as the growth of the time steps from 10 to 25. Different time steps capture varying temporal features, leading to differences in test results. Fewer time steps could cause the excessive concentration of information and then could not extract enough event features, while the larger time steps may extract superfluous information and could not capture the key information.

### 4.4 Comparison with other methods.

We conduct a performance comparison on the CIFAR10-DVS dataset to explore the efficiency of the proposed model. The compared models are as follows: The HOTS method proposed in [29] employs spatio-temporal features called temporal surfaces to create an event-based hierarchical pattern recognition architecture, we adopt three layers of time surface prototypes in the experiment. The article [30] represents the visual information captured by DVS as a stream of asynchronous pixel addresses (events) representing relative intensity changes at those locations, and uses a random ferns classifier to classify the stream of pixel events. The network structure consists of the following three modules: a motion detector, a bank of binary feature extractors, and a Random Ferns classifier. The number of ferns for CIFAR10-DVS is (50, 14), and there are 15,000 segment events with a patch size range between 12 and 20. The paper [31] focuses on visual attention, a specific feature of vision, and proposes two attention models for event-based vision. We adopt the p-N-DRAW recognition network, using pLSTM layers as encoders. The paper [32] introduces a novel event-based feature representation and a new machine learning architecture. And use the local memory unit to efficiently utilize past-time information and build a robust event-based representation. In the experiment, a temporal surface representation method with local memory was used to divide all event points within a spatial region into different fixed cells. Within each cell, all event points are accumulated to generate a temporal surface, obtaining features. Classification is achieved using an SVM (Support Vector Machine). The paper [33] proposes a neuron normalization technique to tune neural selectivity and develops a direct learning algorithm for deep SNNs.The network architecture employed in [33] is as follows: LIAF Blocks and Dense Blocks connected end-to-end.LIAF Block has a sequential structure, consisting of ConvLIAF, TD-layer-normalization, TD-activation (ReLU) and TD-AvgPooling, where TD refers to the time-distributed operation.

A detailed comparison of SNN based models for the benchmark is shown in Table 5, the proposed SRNN-CBAM achieves competitive performance among those models. Our proposed SRNN-CBAM method performs better than the LIF-NET model in [33]. Although the classification accuracy of SRNN-CBAM is lower than LIAF-Net [33], the adaptive memory invoking should be emphasized for the further application of spatiotemporal pattern processing.

# 5 Conclusion

Different from some classical spiking based models which aim at static pattern recognition, this paper utilized the dynamic spatio-temporal information processing ability of SNNs on event data. This paper proposes a brand new spiking recurrent neural network model with spiking convolutional block attention mechanism (SRNN-SCBAM), which can invoke the temporal history information in spatial and temporal features adaptively. It has prior advantages in efficient memory calling and irrelevant history information elimination. The experiment results show that the proposed model can achieve competitive performance among other state-of-the-art models, and the attention component captures meaningful features according to the sensitivity of the output to the input in the end-to-end training process.

At the same time, we believe that the proposed Spiking ConvLSTM is good at different time-series data and applications, such as object tracking, and it is worth noting that the proposed Spiking ConvLSTM needs the spike-based input, hence, when facing other time-series data, the input data should firstly be encoded into spike trains.

In addition, it is worth noting that we take a further step in processing efficient memory invoking and preserving of spiking neural networks, which has promising potential in processing and memorizing spatio-temporal patterns. This paper rethinks the role that attention plays in spiking recurrent neural networks, which not only provides a new way of constructing an efficient model for dynamic event stream data processing but more importantly that the proposed model could mimic the mechanism of potential information processing in neural circuits. Remember what should be remembered, and forget what should be forgotten.

# 6 Acknowledgement

This work was supported in part by National Key Research and Development Program of China (2021ZD0109803), National Natural Science Foundation of China (NSFC No.62206037 and No. 62306274), the China Postdoctoral Science Foundation under Grant No. 2023M733067 and No. 2023T160567, and the Fundamental Research Funds for the Central Universities (DUT21RC(3)091)

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
