# OpenReview forum: "Enhancing Adaptive History Reserving by Spiking Convolutional Block Attention Module in Recurrent Neural Networks"
_NeurIPS.cc/2023/Conference — NeurIPS 2023 poster_

### Official Review · Reviewer_NNKj · 2023-06-26

**Soundness:** 3 good
**Presentation:** 4 excellent
**Contribution:** 3 good
**Rating:** 7
**Confidence:** 5

**Summary:**

The present study introduces a novel model of spiking recurrent neural networks, which incorporates a spike convolutional block attention mechanism. This model is referred to as SRNN-SCBAM. The primary objective is to effectively incorporate historical information into the spatial and temporal characteristics of spatiotemporal patterns, resulting in improved memory retrieval and elimination of redundant historical data. The efficacy of the proposed model in leveraging historical information and attaining high precision has been validated through experiments conducted on DVS128-Gesture datasets.

**Strengths:**

The practical issue of invoking adaptive memory in spiking recurrent neural networks has been addressed in this paper. The primary objective is to incorporate a Spiking Convolutional Block Attention Module into the gating computation of Spiking ConvLSTM networks. This would enable the selective activation of historical information and the elimination of superfluous history during the training phase. This will provide the iterative calculation with a favorable initial state. The motivation is interesting and the idea of utilizing a spiking convolutional block attention mechanism to acquire historical data in gating computation appears intriguing.

**Weaknesses:**

1.More ablations of the different gates being open or closed in Table 1 are required because the influence of ‘i’ is unclear. And if there exist the adaptive parameters to control the possibility of gate open, could it be better for RSNN to achieve better performance?
2.There is a typo in Figure 4.

**Questions:**

1.When encounter a new dataset, how the model works and how the architecture should be designed? I don't see discussions of this part in the paper.
2.More ablations of the different gates being open or closed in Table 1 are required because the influence of ‘i’ is unclear. And if there exist the adaptive parameters to control the possibility of gate open, could it be better for RSNN to achieve better performance?

**Limitations:**

The limitations are the same as the weakness.

---

> ### Author Rebuttal · Authors · 2023-08-09
>
> We appreciate the reviewer's thoughtful comments and insights regarding our article proposing the SRNN-SCBAM model. We would like to address the points raised and provide further clarification on certain aspects.
>
>
> **W1:**"More ablations of the different gates being open or closed in Table 1 "?
>
> **A:**As you suggested, we conduct additional ablation experiments in our study to further explore the effects of different open or closed gates as shown in Table 1. In our subsequent work, we will also investigate the potential of utilizing adaptive parameters or dynamic sub-networks to control gate openings and close.
>
> |    Method | f | i | o | Accuracy |
> |-----------|---|---|---|-----|
> | RSNN-SCBAM | 0 | 1 | 0 | 91.3|
> | RSNN-SCBAM | 1 | 1 | 0 | 92.0|
>
>
> We will incorporate all feedback and correct all typos and incorporate suggestions.

---

> > ### Comment · Reviewer_NNKj · 2023-08-21
> > **Thanks for the response**
> >
> > I would like to thank for the response. My concerns have been addressed.

---

### Official Review · Reviewer_TiEP · 2023-06-29

**Soundness:** 3 good
**Presentation:** 3 good
**Contribution:** 3 good
**Rating:** 7
**Confidence:** 5

**Summary:**

This study proposed the spiking recurrent neural networks model with a spiking convolutional block attention module component (SRNN-SCBAM). The proposed model invokes the historic information in spatial and temporal channels adaptively through spiking CBAM, which brings the advantages of efficient memory calling and history redundancy elimination. The experimental results show that the proposed SRNN-SCBAM model makes better use of the historic information in spatial and temporal dimensions with less memory space, and achieves higher accuracy compared to other models.

**Strengths:**

1. The learning algorithm for recurrent SNNs is simple and easy-to-understand.
2. Well-thought-through reuse of standard modeling and algorithmic components to solve the memory calling and history redundancy elimination in recurrent SNNs.

**Weaknesses:**

Although the motivation is reasonable and the proposed methods are simple and easy to implement, the experimental results could be supplemented to indicate the advantages of the proposed model.

**Questions:**

1. Table 2: What is the network structure in [23]? The network structures should be presented more clearly to show the components in the network models.
2. Why the performance of Spiking ConvLSTM with Spiking CBAM is quite higher than that with CBAM in Table 1?
3. What is the parameter of alpha used in Equation (7)?
4. It seems the proposed Spiking ConvLSTM is suitable for long-term time-series datasets due to its adaptive memory maintenance. Could the proposed model be applied to other time-series datasets?

**Limitations:**

Some details of training the Recurrent SNNs are missing in the paper (only mentioned by a few sentences). It would be better to also include those settings in the appendix.

---

> ### Author Rebuttal · Authors · 2023-08-09
>
> **Q1:**"Table 2: What is the network structure in [23]? "
>
> **A:**The network architecture employed in Table 2 [23] is as follows: 128C3(Encoding)-128C3-AP2-128C3-256C3-AP2-1024FC-Voting.
> We supplement the network structures in Table 2 as follows:
> Table 2 [19] adopts three layers of time surface prototypes.
> Table 2 [20] consists of the following three modules: a motion detector, a bank of binary feature extractors, and a Random Ferns classifier. The number of ferns for CIFAR10-DVS is (50, 14), and there are 15,000 segment events with a patch size range between 12 and 20.
> Table 2 [21] Table 2 [21] adopts the p-N-DRAW recognition network, using pLSTM layers as encoders.
> Table 2 [22] employs the temporal surface representation method with local memory, which divides all event points within a spatial region into different fixed cells. Within each cell, all event points are accumulated to generate a temporal surface, obtaining features. Classification is achieved using an SVM (Support Vector Machine).
>
> **Q2:**"Why the performance of Spiking ConvLSTM with Spiking CBAM is quite higher than that with CBAM in Table 1?
>
> **A:**Firstly, the transmission in SNN  is in the form of spike signals. The compatibility between ANN's CBAM and RSNN might not be particularly suitable to process event-based data. Secondly, in spiking attention, the MultiStepLIFNode is utilized, which leads to the higher performance of Spiking ConvLSTM with Spiking CBAM compared to using conventional CBAM.
>
> **Q3:**"What is the parameter of alpha used in Equation (7)?"
>
> **A:**$\alpha$ is the parameter that controls the smoothness of the gradient during backpropagation for surrogate gradient learning process, which is set to be 4.0 in this paper.
> Currently, it appears that the Spiking ConvLSTM could be applicable to other DVS (Dynamic Vision Sensor) datasets. Exploring its applicability to other time-series datasets is also a focus of our future work.
>
>
> **Q4:**"It seems the proposed Spiking ConvLSTM is suitable for long-term time-series datasets due to its adaptive memory maintenance. Could the proposed model be applied to other time-series datasets?"
>
> **A:**Yes, the proposed model can be applied to other time-series datasets. The proposed Spiking ConvLSTM is good at the different time-series data and applications, such as the object tracking, and it is worth noting that the  proposed Spiking ConvLSTM needs the spike-based input, hence, when facing other time-series data, the input data should firstly be encoded into spike trains.

---

> > ### Comment · Reviewer_TiEP · 2023-08-11
> >
> > I have read the response. All my concerns have been well addressed.

---

### Official Review · Reviewer_KuzU · 2023-07-03

**Soundness:** 2 fair
**Presentation:** 1 poor
**Contribution:** 1 poor
**Rating:** 3
**Confidence:** 5

**Summary:**

The authors proposed a Recurrent spiking neural network (RSNN). The essential component of the proposed RSNN is a spiking conv block attention module (SCBAM), which contains channel and spatial attention blocks. The proposed method is validated with classification tasks on CIFAR10-DVS and DVS128 gesture datasets.

**Strengths:**

N/A

**Weaknesses:**

-1. The writing of the submission should be dramatically improved. Many things are hard to follow. In lines 242-243: "Although the ...is lower than LIAF-Net, ...". In Table 2, [23] provides lower accuracy than the proposed method. Therefore, I cannot really trust the provided experimental results.

-2. lines 114-115: "Spiking Convlstm is more suitable for removing  facial expression data...." Why?

-3. SNN is different from CNN, which can process temporal information without any treatment, especially since many works have proved that, such as [R-1], [R-2]. The process of membrane potential release, accumulation, and triggering spiking is a natural temporal extractor. I do not think using RNN-based architecture is a meaningful way to go in the SNN domain. At the minimum, the authors should provide experimental results of the comparison with these methods, such as [R-1] and [R-2].

-4. How does the potential threshold of LIF impact the performance of the proposed methods?

-5. How does the number of LSTM steps impact the performance of the proposed methods?

[R-1]: Biologically Inspired Dynamic Thresholds for Spiking neural networks.

[R-2]: Spiking Transformers for Event-based Single object Tracking

**Questions:**

Please see the weaknesses section.

**Limitations:**

No limitation is provided.

From my perspective, the work follows the thoughts of ANN without an insightful understanding of SNN, especially the temporal processing power of SNNs. SNNs offer compelling temporal feature extraction capability without any special treatment, evidenced by many works already.

In addition, I did not see the novelty of the proposed approach from either ANN or SNN. Therefore, I think the submission is way below the bar of NeurIPS.

---

> ### Author Rebuttal · Authors · 2023-08-09
>
> Dear Reviewer KuzU,
>
> Thank you for the thorough review and constructive criticism. There exist some misunderstanding about our paper, we hope the following the clarification would solve the proposed problems.
>
> **W1:**"the mismatch about the description about comparison with LIAF-NET" ?
>
> **A:**Thanks for pointing that. Actually, [23] contains two different model settings of LIF and LIAF with two different accuracies of 63.53% and 70.40% on CIFAR10-DVS dataset, respectively. Because the LIF model is employed in this study, we show the former result with 63.53% accuracy in Table 2 for direct comparison with our study. Meanwhile, we also refer the latter results of 70.40% in lines 242-243, with the desctiption of " Although the classification accuracy of SRNN-CBAM is lower than LIAF-Net [23], the adaptive memory invoking should be emphasized for the further application of spatiotemporal pattern processing." to show the advantages and disadvantage of the proposed RSNN to illustrate the whole view of our study.
>
> We would rewrite the corresponding description to make it clear, by adding both the results of LIF-NET and LIAE-NET in [23] into Table 2 and supplement the description of "Our proposed SRNN-CBAM method performs better than the LIF-NET model in [23]."
>
> **W2:**"Spiking ConvLSTM is more suitable for removing facial expression data"?
>
> **A:**Here It means that the Spiking ConvLSTM is more suitable for spike-based coarse-grained data instead of the  fine-grained images such as the facial expression data. In detail, on DVSGesture dataset, each sample is captured by the dynamic vision sensor (DVS), and the facial expression during finishing each gesture process is removed because of the event-driven response property of DVS. Hence, the Spiking ConvLSTM is more suitable to the DVS data instead of the pixel-level fine-grained images.
>
> Thanks for pointing that, we would change the description to "Spiking ConvLSTM is more suitable for processing the dynamic spatiotemporal patterns, such as the gesture recognition with the clear gesture moving trajectories but without preserving the details of fine-grained images with high resolution pixel.
> "
>
> **W3 and W4:**"comparison with [R-1] and [R-2]" ? "How does the potential threshold of LIF impact the performance of the proposed methods?"
>
>
> **A:**: Thanks for your suggestion. Here we conducted ablation experiments about the dynamic threshold during training on the DVSGesture dataset, although we could not compare the spiking Transformer in [R-2] for its object tracking framework can not directly applied into our application scene. Two different dynamic threshold methods in SNNs with LIFNode and MultiStepLIFNode are employed in our RSNN-SCBAM model.  We make the thresholds of MultiStepLIFNode and LIFNode trainable parameters and conducted experiments to validate their impact. The results are as follows:
>
> |    Method | Node_ threshold | Acc (%)|
> |---------------|-------------|-------------|
> | RSNN-SCBAM	| LIFNode		| 91.3        |
> | RSNN-SCBAM	| MultiStepLIFNode        | 89.9          |
>
> However, the current results do not show any significant improvements with the dynamic threshold. This threshold adjustment could make the model relatively intricate, involving an increased number of parameters and non-linear operations. If the model's parameters or network structure are not adjusted correctly, it might lead to suboptimal model performance. Additionally, SNNs might demand a larger amount of training data and an extended training time for effective learning and optimization.
>
> [1] Dayan P, Abbott L. Computational neuroscience: Theoretical neuroscience: Computational and mathematical modeling of neural systems[M]. Cambridge: MIT Press, 2001: 162-166.
> [2] Brunel N, Latham P E. Firing rate of the noisy quadratic integrate-and-fire neuron[J]. Neural Computation, 2003, 15(10): 2281-2306.
> [3] Pellegrini T, Zimmer R, Masquelier T. Low-activity supervised convolutional spiking neural networks applied to speech commands recognition[C]//2021 IEEE Spoken Language Technology Workshop (SLT). IEEE, 2021: 97-103.
>
> **W5:**"How does the number steps impact the performance of the proposed methods?"
>
> **A:**Regarding the variation in LSTM time steps, we conduct experiments using additional time step values of 10, 15, and 25 on the DVSGesture dataset. As shown in the following table, the results shows that  the time step of 20 is the optimal choice. This could be attributed to the nature of the DVSGesture data, where gestures are repeated by individuals at specific time intervals. The accuracis of the RSNN-CSAM increase first and then reduce as the growth of the time steps from 10 to 25. Different time steps capture varying temporal features, leading to differences in test results. Less time steps could cause the excessive concentration of information and then could not extract enough event feature, while the larger time steps may extract superfluous information and could not capture the key information.
>
>
> |    Method | Time step | Accuracy (%) |
> |---------------|-------------|-------------|
> | RSNN-SCBAM	| 10		| 89.2        |
> | RSNN-SCBAM	| 15        | 92.4          |
> | RSNN-SCBAM	| 25        | 90.3        |

---

### Official Review · Reviewer_9YjJ · 2023-07-05

**Soundness:** 4 excellent
**Presentation:** 3 good
**Contribution:** 4 excellent
**Rating:** 7
**Confidence:** 5

**Summary:**

This article proposes a spike recurrent neural network model with a spiking convolutional block attention mechanism, called SRNN-SCBAM. Its main idea is to adaptively call historical information in the spatio-temporal features of the spatio-temporal pattern, which has advantages in efficient memory calls and eliminating historical redundancy. The experiment on the DVS128 gesture dataset is conducted to verify the effectiveness of the model in utilizing adaptive historical information. The idea of the article is clear, the textual description is clear, and examples are provided to illustrate results.

**Strengths:**

1.The paper is easy to read, and generally well written.
2.The proposed model of RSNN is simple but effective, the motivation is clear.
3.Experimental evaluations were conducted on two different neuromorphic datasets to demonstrate the performance improvement with the proposed SCBAM under different settings.

**Weaknesses:**

There is a lack of explanation for some parameter settings. In addition, there are some shortcomings in the experimental aspect of the paper. The experimental results mentioned in the article on the Cifar10-DVS generally can be supplemented with appropriate visualization results.

**Questions:**

1.The parameter setting about the time constant in the first paragraph of section 2.4 should be explained in detail.
2.In addition, there are some shortcomings in the experimental aspect of the paper. The visualization experimental results mentioned in the article on the Cifar10-DVS can be supplemented with appropriate experiments.

**Limitations:**

1.The parameter should be explained further.
2.More visualization results could be added.

---

> ### Author Rebuttal · Authors · 2023-08-09
>
> Dear Reviewer 9YjJ,
>
> We really appreciate your insightful comments and feedback. We addressed your questions below. We also revised our paper accordingly.
>
>
> **Q1:** "The parameter setting."
>
>  **A:**Thanks for the suggestion. All the parameter settings are listed in the following table.
> Together with the description of section 4.1, the following parameters' setting will be added to the revision: The integration time constants $\tau$ of LIF neurons are set to be 4.0.
> $\alpha$ is the parameter that controls the smoothness of the gradient during backpropagation for surrogate gradient learning process, which is set to be 4.0 in this paper.
> $u_{reset}$ is the reset voltage of the neuron and set to be 0. The above content would be added to the revision.
>
> |    Parameters | DVS Gesture | Cifar10-DVS |
> |---------------|-------------|-------------|
> | Epoch         | 200        | 1024        |
> | Batchsize     | 32          | 32          |
> | LIF, $	au$    | 2.0         | 2.0         |
> | T             | 20          | 20          |
> | $\alpha$      | 4.0         | 4.0         |
> | $u_{reset}$   | 0           | 0           |
>
> **Q2:**"the visualization results on CIFAR10-DVS".
>
> **A:**Thanks for the suggestion, we have supplemented appropriate visual results on CIFAR10-DVS dataset, as shown in the one-page PDF rebuttal file. In detail, we visualize the extracted features at several time stps of the proposed RSNNs with and without SCBAM.  It shows that the RSNN module with SCBAM captures information as anticipated, exhibiting strong sparsity across both temporal and spatial dimensions throughout the entire time steps while the RSNN module without SCBAM only extracts limited features which may lose a significant amount of crucial information.
>
> Thanks again for all of your valuable feedback, suggestions, and questions.

---

### Author Rebuttal · Authors · 2023-08-09

We thank the reviewers for their helpful feedback. We are inspired by the fact that they have found our motivation clear [Reviewer 9YjJ], reasonable [Reviewer TiEP] and our proposed approach simple but effective [Reviewer 9YjJ] and novel [Reviewer NNKj]. We appreciate that  [Reviewer TiEP] expresses agreement, stating, "This would enable the selective activation of historical information and the elimination of superfluous history during the training phase. This will provide the iterative calculation with a favorable initial state." We are delighted that everyone agrees our experimental evaluation is clear, and the proposed model is highly suitable for handling spatiotemporal data. We are also pleased that the reviewers found the paper to be well-presented and well-organized [Reviewer 9YjJ], and the ideas presented to be intriguing [Reviewer TiEP].

Here, we emphasize the contributions in this paper:

We introduce the Spiking Recurrent Neural Networks model with the Spiking Convolutional Block Attention Module (SRNN-SCBAM), aiming to leverage both spatial and temporal features of spatio-temporal patterns effectively. The SRNN-SCBAM model utilizes the spiking CBAM component to adaptively incorporate historical information from spatial and temporal channels, thereby benefiting from efficient memory retrieval and eliminating redundancy in historical data.
To validate the model's efficacy, extensive experiments are conducted on CIFAR10-DVS and DVS128-Gesture datasets. The ablation study and comparison results demonstrate that our proposed model achieves competitive performance among the existing RSNN models.

Thanks to the reviews’ suggestions on experimental description [Reviewer 9YjJ], experimental settings [Reviewer KuzU, Reviewer TiEP] and biological plausibility [Reviewer TiEP]. We have noticed the flaws in the origin description from the constructive feedback from the reviewers and have prepared an updated version of the manuscripts. Here we will roughly summarize the changes made. For a more detailed reply and explanation, please refer to the individual responses below.

**About the nolvety:** As [Reviewer TiEP] expresses agreement, stating, "Well-thought-through reuse of standard modeling and algorithmic components to solve the memory calling and history redundancy elimination in recurrent SNNs". We think our study is not the simple combinnation of the existing methods within the same paradigm.

**The comparison with SNNs with learnable threshold**We added the experimental results with the dynamic thresholds of MultiStepLIFNode and LIFNode trainable.

**The parameter settings**We explained all the parameter settings and the network structures.

**More visualization results**We added the visualization results in the one-page PDF rebuttal file.

We will incorporate all feedback and correct all typos and incorporate suggestions. We appreciate the reviewer’s thoughtful comments and insights regarding our article proposing the SRNN-SCBAM model. We would like to address the points raised and provide further clarification on certain aspects.

Thanks again for all reviewers, authors of paper 2180.

---

### Decision · Program_Chairs · 2023-09-21

**Decision:**

Accept (poster)

**Comment:**

This paper got mixed recommendations, 3 accept and 1 reject. On the positive side, most of the reviewers agree that this paper presents clear motivation, novel method and model design and solid experiment results. On the negative side, one reviewer raised several questions on the detailed hyperparameter choice and their impact and also raised concerns on the novelty. The authors have provided detailed response to these questions. The AC has read all the reviews and responses from the authors. The authors have provided detailed answers to the questions that, to the AC, have addressed the questions. Therefore, the AC agree with most of the reviewers and recommend accept for this paper.